# Development of New Antimicrobial Peptides by Directional Selection

**DOI:** 10.3390/antibiotics14111120

**Published:** 2025-11-06

**Authors:** Ekaterina Grafskaia, Pavel Bobrovsky, Daria Kharlampieva, Ksenia Brovina, Maria Serebrennikova, Sabina Alieva, Oksana Selezneva, Ekaterina Bessonova, Vassili Lazarev, Valentin Manuvera

**Affiliations:** 1Laboratory of Genetic Engineering, Lopukhin Federal Research and Clinical Center of Physical-Chemical Medicine of Federal Medical Biological Agency, Moscow 119435, Russia; grafskayacath@gmail.com (E.G.); harlampieva_d@mail.ru (D.K.); ksenia.brovina33@yandex.ru (K.B.); maria.serebrennikova.msu@gmail.com (M.S.); stadashi6@gmail.com (S.A.); oks.selezneva36@gmail.com (O.S.); bes9122007@yandex.ru (E.B.); lazarev@rcpcm.org (V.L.); vmanuvera@yandex.ru (V.M.); 2Moscow Center for Advanced Studies, 20, Kulakova Str., Moscow 123592, Russia

**Keywords:** antimicrobial peptide, antibacterial activity, mutagenesis, library, Hm-AMP2, melittin, cecropin, expression system

## Abstract

**Background/Objectives**: The global rise in antibiotic resistance necessitates the development of novel antimicrobial agents. Antimicrobial peptides (AMPs), key components of innate immunity, are promising candidates. This study aimed to develop novel therapeutic peptides with enhanced properties through the mutagenesis of natural AMPs and high-throughput screening. **Methods**: We constructed mutant libraries of three broad-spectrum AMPs—melittin, cecropin, and Hm-AMP2—using mutagenesis with partially degenerate oligonucleotides. Libraries were expressed in *Escherichia coli*, and antimicrobial activity was assessed through bacterial growth kinetics and droplet serial dilution assays. Candidate molecules were identified by DNA sequencing, and the most promising variants were chemically synthesized. Antimicrobial activity was determined by minimal inhibitory concentration (MIC) against *E. coli* and *Bacillus subtilis*, while cytotoxicity was evaluated in human Expi293F cells (IC_90_) viability. The therapeutic index was calculated as the ratio of an AMP’s cytotoxic concentration to its effective antimicrobial concentration. **Results**: Mutant forms of melittin (MR1P7, MR1P8) showed significantly reduced cytotoxicity while retaining antimicrobial activity. Cecropin mutants exhibited reduced efficacy against *E. coli*, but variants CR2P2, CR2P7, and CR2P8 gained activity against Gram-positive bacteria. Mutagenesis of Hm-AMP2 generally decreased activity against *E. coli*, though two variants (A2R1P5 and A2R3P6) showed retained or enhanced efficacy against *B. subtilis* while maintaining low cytotoxicity. **Conclusions**: The proposed strategy successfully generated peptides with improved therapeutic profiles, including reduced toxicity or a broader spectrum of antimicrobial activity, despite not improving all parameters. This approach enables the discovery of novel bioactive peptides to combat antibiotic-resistant pathogens.

## 1. Introduction

In the middle of the 20th century, antibiotics were introduced into medical practice as powerful new agents of combating bacterial infections, and they rapidly gained widespread use. However, the extensive application of these drugs soon led to the emergence of antibiotic-resistant bacterial strains. Over time, the problem of drug resistance in pathogenic bacteria has only intensified [1,2]. This situation underscores the urgent need for continuous discovery and development of novel antimicrobial agents.

In this pursuit, increasing attention has been directed toward natural defence systems against infections. One of the most common mechanisms used by animals to fight pathogens is the production of antimicrobial peptides (AMPs) [3]. To date, a large number of AMPs from diverse natural sources have been described, and several have been evaluated for medical applications [4,5,6,7]. Nevertheless, natural AMPs face several limitations that hinder their practical use. Some of these constraints arise from the shortcomings of genome- and proteome-based discovery methods, which fail to capture the full diversity of possible AMPs [8,9,10,11,12]. Others are intrinsic to the peptides themselves. For instance, the majority of known AMPs have been characterized in invertebrates [13,14,15]. As a result, many natural peptides evolved to function in environments quite different from those found in the human body [16,17]. Physicochemical conditions in a leech’s mucous cocoon or marine cnidarians differ greatly from human physiological settings. Moreover, the spectrum of microorganisms targeted by innate immunity of arthropods, worms, or cnidarians is distinct from that which threatens human health [18,19]. Natural AMPs must also maintain a delicate balance: eliminating pathogens while preserving the stable microbiome [20,21]. Consequently, natural AMPs may represent not the most effective antimicrobial agents, but rather compromise solutions adapted to ecological balance.

Despite these challenges, AMPs are regarded as promising adjuncts to antibiotics [16], and the search for new peptides remains an important goal. However, the initial enthusiasm has not translated into the widespread clinical application of AMP-based therapeutics, primarily due to a mismatch between the properties of natural AMPs and the requirements for practical use. One way to overcome these barriers is through the rational design of AMPs. Currently, in silico approaches are commonly employed, where peptide structures are first computationally designed, then synthesized and experimentally validated [22,23,24,25].

In the present study, we applied a fundamentally different strategy (Figure 1). Instead of computational design, we generated libraries of expression plasmids encoding random mutant variants of well-characterized AMPs. The antimicrobial activity of the mutant peptides was assessed using a novel primary screening method based on the expression of recombinant AMP genes in *Escherichia coli* [26]. Briefly, the plasmid libraries were used to transform *E. coli* cells, which were then induced to express the recombinant AMPs. Clones exhibiting the strongest growth inhibition upon induction were selected for the next rounds of mutagenesis and screening. In this way, we carried out large-scale mutagenesis of known AMPs while retaining only those variants that preserved antimicrobial activity. Finally, the selected mutant peptides were synthesized and characterized.

As model AMPs, we employed three well-studied peptides with broad antimicrobial activity: melittin from the honeybee [27], Hm-AMP2 from the medicinal leech [28], and cecropin from *Ascaris suum* [29]. We suggest that this approach can provide new peptide variants with enhanced antimicrobial activity, reduced cytotoxicity toward human cells, or specificity against particular bacterial groups.

## 2. Results

### 2.1. Determination of Mutation Frequency in Plasmid Libraries

Mutant plasmid libraries for each AMPs were constructed based on the pET22bStop plasmid. In this modified version of pET22b, the native DNA region encoding stop codon was extended to form a stop codon that functions in all three reading frames. Thus, in the event of a frameshift, it would not result in a mutant peptide longer than the original. Partially degenerate oligonucleotides were synthesized using phosphoramidite mixtures, each containing 88% vol. of the main phosphoramidite and 4% vol. each of the remaining three. The assembled plasmid libraries were transformed into *E. coli* Top10 cells. The mutation frequency for each plasmid library was determined by PCR analysis of 96 colonies. Using T7/T7t primers and cell suspensions of these colonies as a template, amplicons were obtained and subsequently sequenced. The observed mutation frequency in the AMP encoding genes was generally consistent with the predicted values but showed a bias toward higher numbers (Figure 2). This is primarily due to random errors during oligonucleotide synthesis, and not from the deliberate addition of bases to the reagents. For instance, a significant number of DNA fragments contained single-nucleotide deletions. Analysis of clones constructed using conventional (non-degenerate) oligonucleotides also revealed the presence of random substitutions and deletions in some clones.

### 2.2. Testing the Antimicrobial Activity of Libraries Encoding Mutant AMP Variants

We performed a sequential three-stage screening of libraries encoding mutant variants of Hm-AMP2. Experiments were conducted using *E. coli* BL21(DE3)gold, and antimicrobial activity was quantitatively assessed using the R_OD_ metric (ratio of optical density in uninduced vs. induced cultures), which served as an objective measure of bactericidal activity. We screened at least 192 colonies from each library. At each subsequent stage, the most active peptide selected from the previous screening round was used as a backbone for the next mutant plasmid library construction. The selected clone was also included in assays as an additional control.

In the first round of screening, initial evaluation identified 22 Hm-AMP2 mutants with substantial variability in antimicrobial activity. Statistical analysis revealed a significant difference in activity for only 5 tested peptides compared to native Hm-AMP2 (Figure 3A). Among them, clone A2R1P5 exhibited high antimicrobial activity (R_OD_ 6.4 ± 0.7, *p* = 0.03) and was selected for the second round of mutagenesis. The results were further verified using a droplet serial dilution assay (DPSD). As described in Section 4.5, tenfold serial dilutions of the cell suspensions were prepared. For each dilution of each clone, 10 µL droplets were placed on agar plates and allowed to absorb. The first dilution at which no colonies appeared in the spot was recorded. In this analysis, peptide A2R1P5 (DPSD = 10^3^) also demonstrated superior activity compared to native Hm-AMP2 (DPSD = 10^4^) (Figure 3A).

The second round of mutagenesis and library analysis yielded 10 new mutant variants. Kinetic experiments revealed considerable variation in antimicrobial activity among these peptides. The five most active peptides were selected for further analysis. Statistical evaluation identified two of the most promising variants: peptide A2R2P3 (R_OD_ 5.4 ± 0.6, *p* = 0.0014) and peptide A2R2P5 (R_OD_ 5.7 ± 0.4, *p* = 0.005) (Figure 3B). Both peptides outperformed the control (A2R1P5) in droplet serial dilution assays and were selected for further optimization.

In the final stage, libraries derived from both A2R2P3 and A2R2P5 were generated and analyzed. Eighteen new mutant variants were identified from these libraries. Screening results revealed a clear trend: all five peptides that showed statistically significant superiority over the control belonged to the A2R2P5 mutant lineage (Figure 3C). No variants from the A2R2P3 lineage significantly exceeded the activity (R_OD_) of the parent peptide A2R2P3. Droplet serial dilution assays further confirmed the enhanced activity of peptides A2R3P4 and A2R3P5.

A similar screening was performed for libraries of melittin mutant variants. Due to the high intrinsic activity of melittin, we were unable to identify mutant variants that were significantly more active than the native peptide. Out of 174 tested peptides, nine were selected that exhibited activity comparable to native melittin (Figure 4). Although the control peptide exhibited a wide spread of R_OD_ values, the medians for all selected peptides were within its interquartile range (25th–75th percentiles); so, we could not reject the null hypothesis of equivalent activity. The five most active peptides were chosen, among which clone MR1P5 showed the highest antimicrobial activity (R_OD_ 32.1 ± 3.2). Given melittin’s considerable cytotoxicity, only one round of mutagenesis was performed, as further modification would not have allowed for a reliable assessment of the effects of mutations on the activity of the mutant variants.

Next, we screened libraries of cecropin mutant variants. Similar to the melittin mutants, we were unable to identify peptides with activity significantly higher than native cecropin based on statistical analysis. However, we were able to eliminate variants with reduced activity following mutagenesis (Figure 5A). During the first round of mutagenesis, peptide CR1P5 was selected as the most active variant, with a median R_OD_ higher than the control (R_OD_ 7.0 ± 2.1). This sequence was then used for the second round of mutagenesis. Analysis of the second-round libraries, comprising 161 sequences, led to the selection of seven promising variants. Among these, five peptides exhibited the highest activity (Figure 5B). Subsequent statistical analysis identified one particularly promising variant: peptide CR2P3 (R_OD_ 33.9 ± 0.3, *p* = 0.01).

### 2.3. Influence of the Amino Acid Substitutions on Antimicrobial Peptide Activity

To identify critical amino acid positions or patterns governing antimicrobial activity, a series of experiments was performed to assess plasmid libraries encoding variants of the HmAmp2, cecropin, and melittin peptides. Transformant growth analysis (Section 2.2) allowed for determination of the antimicrobial activity level for each peptide variant. To remove artifacts, duplicate amino acid sequences were excluded from the dataset. For peptides present in multiple replicates, R_OD_ values were averaged.

All unique peptide sequences were divided into two groups based on their activity relative to the reference antimicrobial peptide (Hm-AMP2, melittin, or cecropin). The “high” group included peptides with higher activity than the original peptide, while the “low” group contained peptides with lower activity. This approach enabled a comparative analysis of structural features associated with increased or decreased antimicrobial activity.

Amino acid residues were then classified into four major groups according to their physicochemical properties. The effect of substitutions on class changes (e.g., polar → nonpolar) was evaluated for each residue. For each position in the peptide sequence, the frequency of such substitutions was calculated, and frequency histograms were generated (Figure 6). This analysis revealed patterns in the distribution of critical substitutions along the peptide chain. In all cases, the first amino acid in the peptide remained unchanged, as nucleotides encoding it were synthesized using individual phosphoramidites rather than mixtures.

In the first round of Hm-AMP2 mutagenesis, several critical positions were identified where substitutions positively affected peptide activity (Figure 6A–C). Positions L7 and Y11 were notable: class-changing mutations at position 7 occurred only in low-activity variants, whereas substitutions at position 11 occurred exclusively in high-activity variants. At position 3, replacement of R with a residue from a different class predominantly led to increased activity. In peptides selected for the second and third rounds of mutagenesis, R3G substitutions were observed. Changes at this position decreased activity in Hm-AMP2 variants. Substitutions at position 7 negatively affected activity: L7W replaced a nonpolar residue with another nonpolar residue, while L7C (nonpolar → polar) in clone A2R2P5 resulted in lower activity compared to A2R2P3 (Figure 3). Analysis of mutants in this peptide indicated that certain substitutions at this position could again enhance activity. Positions 4, 8, 9, and 12 appeared critical; they remained unchanged in all selected peptides. Their importance became more pronounced in the second and third rounds of mutagenesis compared to the first.

For the melittin library, critical positions were identified where amino acid substitutions either significantly increased or decreased antimicrobial activity (Figure 6D). Some mutations occurred exclusively in inactive peptides—positions L6 and L16. Substitutions at L9 and L13 also predominantly resulted in loss of activity, with mutation frequencies of 2% in active peptides versus 17% and 18% in inactive peptides, respectively. Residues at positions 2–5 and 22–26 appeared to have minimal impact on melittin activity.

Analysis of cecropin mutant libraries revealed numerous critical positions where mutations occurred exclusively in either active or inactive peptides (Figure 6E,F). Positions 2 and 3 were apparently critical. The amino acid class remained unchanged in the second round. During the second round, three peptides retained their length, while selected peptides CR2P4 and CR2P5 differed in length due to synthesis errors in oligonucleotides. All five peptides shared a similar N-terminal sequence of 19 residues. Position R17 was also critical according to first-round data; this residue remained unchanged in all selected second-round peptides.

### 2.4. Testing the Activity of Synthetic Peptides

To evaluate our method’s efficacy, we selected peptides from the plasmid library screening with the highest and lowest antimicrobial activity (R_OD_ values) for each reference AMPs. We successfully synthesized only a subset of peptides, as synthesis failures occurred for highly hydrophobic sequences. For the synthesized peptides, we determined the minimum inhibitory concentration (MIC) against *B. subtilis* (MIC_Bs_) and *E. coli* (MIC_Ec_) and measured cytotoxicity (IC_90_) in Expi293F cells (human embryonic kidney cells) using an MTT assay (Figure 7A–C, Appendix A). The IC_90_ value, representing the concentration of AMP that inhibited 90% of cell growth, was used to quantify its cytotoxicity.

Using the measured MIC and cytotoxicity values, the therapeutic index (TI) for each peptide was calculated. The TI was defined as the ratio of the concentration required for drug efficacy to the concentration causing toxicity to mammalian cells. In our study, it was calculated as the minimum cytotoxic concentration (IC_90_) divided by the maximum MIC value [30]. For all tested peptides, MIC_Ec_ was maintained or increased, indicating preserved or reduced antimicrobial activity against *E. coli*. In contrast, MIC_Bs_ decreased for several variants compared to the original peptide. For cecropin, these were CR2P8, CR2P9, and CR2P2. For Hm-AMP2, variants A2R1P5, A2R2P7, and A2R3P8 showed reduced MIC_Bs_. Mutant forms of melittin did not show a reduction in MIC; however, all exhibited substantially lower cytotoxicity, resulting in an increased calculated TI.

## 3. Discussion

Antimicrobial resistance is a growing threat to global health and the economy, potentially leading to 10 million deaths annually by 2050 [1,2,31]. Its development is promoted by antibiotic overuse, self-medication, and genetic adaptation of microbes [32]. Hospital-acquired infections caused by the group of resistant ESKAPE pathogens pose the greatest threat due to high mortality and treatment costs [33]. Antimicrobial peptides (AMPs) are considered a promising alternative to conventional antibiotics due to their potent efficacy and low potential for cross-resistance [3]. This advantageous profile is underpinned by their fundamental mechanism of action. AMPs initially bind to anionic bacterial surfaces via electrostatic attraction, a step that provides selectivity over mammalian cells [34]. Subsequent bactericidal actions, however, are heterogeneous, involving diverse pathways such as pore formation or detergent-like dissolution of the lipid bilayer, either exclusively or concurrently [35,36]. Two main approaches are used to discover new AMPs. Rational design involves creating new compounds based on templates, physicochemical properties, or de novo design using a machine learning approach [37,38]. Non-rational design relies on high-throughput screening of natural or synthetic libraries (phage, bacterial, or yeast display) [39,40,41,42].

In this study, we proposed a hybrid approach combining the design of new peptides based on known AMPs using mutant libraries with selection in bacterial cells expressing these novel AMPs. Our approach serves as a functional alternative to purely computational design. Unlike in silico methods that predict a structure before synthesis, our method is empirical and functional. It enables the testing of vast libraries of variants directly inside the bacterial cell, selecting them not for predicted stability, but for their actual antimicrobial activity. A key distinction from standard mutagenesis is that we do not merely create random mutants. Instead, we use an iterative process (rounds of mutagenesis and screening) that mimics natural selection, guiding the evolution of the peptides toward preserving or enhancing their function under physiologically relevant conditions (inside the bacterial cell).

To implement this, specifically to compare the properties of mutant AMPs and their selection, it was first necessary to establish a mutagenesis methodology that would provide a predictable level of random mutations in the coding sequences of recombinant genes. We tested several classical methods, including amplification using dITP [43,44]. However, the results were unstable and poorly predictable due to the short sequence lengths, and the mutagenesis was not fully random. Consequently, we used an alternative approach to generate mutant fragment libraries. DNA fragments encoding AMPs were assembled from duplexes of synthetic oligonucleotides with sticky ends complementary to vector ends produced by restriction enzyme digestion. To achieve a uniform, controlled level of nucleotide substitutions, we used mixtures of monomers during oligonucleotide synthesis, as described in Section 4.2. This enabled the generation of plasmid libraries with a controlled level of nucleotide substitutions in the coding regions of recombinant genes, as described in Section 2.1. Sequencing of the libraries confirmed that the mutagenesis level matched the calculated values, with additional consideration for random errors introduced by using non-degenerate oligonucleotides. Thus, the generation of plasmid libraries was successfully achieved.

In the next stage, we screened the antimicrobial activity of the generated plasmid libraries encoding AMPs [26]. Plasmids were used to transform *E. coli* cells, and the resulting clones were transferred into liquid culture in 96-well microplates. After short-term growth, IPTG was added to induce AMP expression. Cultures expressing active AMPs exhibited growth suppression. For quantitative measurement, we used the R_OD_ value, defined as the ratio of the optical density of cultures without inducer to that of cultures with IPTG (Section 4.4). Higher R_OD_ values correspond to stronger bacterial growth suppression upon AMP expression. This screening method is highly efficient, enabling the testing of a large number of sequences with moderate resource use, whereas screening a comparable number of synthetic peptides would be far more labor-intensive and costly [26,45,46]. However, a critical interpretation of the results necessitates consideration of the method’s key limitations. The primary constraint is the unknown AMP expression level in individual clones, which is influenced by gene structure and peptide stability. This issue arises from the fundamental inability to accurately measure the synthesis and intracellular accumulation of AMPs. Direct detection to quantify expression is not feasible. Detection at the RNA level is unreliable because the transcript levels do not reliably correlate with protein levels, and transcripts from mutant libraries exhibit high stochasticity. Similarly, immunodetection is unworkable because high-level expression of active AMPs induces cell death, limiting the peptide available for detection, while the varying intracellular half-lives of different peptides further complicate reliable quantification.

Consequently, the observed changes in growth inhibition (R_OD_) could be a result of either the true intrinsic antimicrobial activity of the peptides or differences in their expression levels. This means the method does not allow for a direct comparison of the relative activity of the peptides or for the determination of quantitative parameters (MIC, IC_90_) based on the screening data alone, as the R_OD_ value reflects the combined effect of peptide expression and function. It is also crucial to note the difference between intracellular accumulation during heterologous expression and the extracellular activity of synthetic peptides.

In our study, some of these limitations were mitigated because the compared peptides were of similar length and primary structure. In this regard, the screening data are of a preliminary nature. The final functional profiles of the selected variants (MIC, IC_90_, TI) were established in subsequent experiments using synthetic peptides in standard extracellular assays. Therefore, while the proposed plasmid-based screening is an effective tool for the primary screening of extensive AMP libraries, its results require mandatory validation by traditional microbiological methods.

Correlation of R_OD_ values, reflecting bacterial growth suppression, with sequencing data allowed for identification of amino acid positions where substitutions statistically led to decreased antimicrobial activity, while substitutions at other positions were associated with increased R_OD_, though this effect was less pronounced (Section 2.3). Due to the mentioned limitations, these data should be interpreted cautiously and in conjunction with more rigorous experiments.

The power of this functional screening approach was demonstrated through the distinct evolutionary paths revealed for our three model AMPs. For Hm-AMP2, three rounds of mutagenesis were performed. The first round used the native Hm-AMP2 sequence, while subsequent rounds used the most active variant from the previous round. For cecropin, two rounds were performed, and for melittin, only one round. The number of rounds was determined based on activity testing results. In Hm-AMP2, variants exceeding the original peptide’s R_OD_ were observed in the first and second rounds. For cecropin, this occurred only in the first round, and no variants exceeded the control in the second round. For melittin, no variants exceeded the native peptide in the first round. Melittin is a small, highly toxic peptide, acting more as a cytolytic toxin than a specialized antimicrobial agent [27,47]. It is likely evolutionarily optimized for maximal membrane disruption, and introduced mutations mainly reduce its toxicity [48]. Consistently, all selected melittin variants exhibited increased IC_90_ (Figure 7A–C, Appendix A), with one variant, MR1P7, showing a tenfold reduction in cytotoxicity while maintaining MIC values. Overall, the increase in IC_90_ for melittin variants was more pronounced than any decrease in MIC, resulting in higher calculated therapeutic index (TI). The native melittin TI was 0.128, while four out of five mutant variants had higher TI, reaching 2 in MR1P8.

In contrast, cecropin is a specialized AMP targeting Gram-negative bacteria [29]. It has low cytotoxicity (IC_90_ = 200), low MIC_Ec_ (0.39 μM), and no toxicity toward *B. subtilis*. Its sequence is slightly longer (31 aa vs. 26 aa for melittin). Among mutant cecropin variants, greater diversity of properties was observed than in melittin. All six tested variants showed reduced MIC_Ec_, but three acquired activity against *B. subtilis*. One variant, CR2P6, lost all activity despite initially high R_OD_ during selection, likely due to differences in intracellular versus extracellular peptide localization. All cecropin variants retained low IC_90_, and TI values were slightly lower than the native peptide, with an expanded Gram-positive activity profile.

The Hm-AMP2 variants showed the most heterogeneous outcomes. Native Hm-AMP2 exhibits activity against both *E. coli* and *B. subtilis*, with low cytotoxicity [28], representing a broad-spectrum evolutionarily optimized AMP. MIC_Ec_ was higher than that of cecropin (3.1 μM vs. 0.39 μM) but Hm-AMP2 retained activity against *B. subtilis* (MIC_Bs_ = 12.5 μM). All Hm-AMP2 variants showed equal or higher MIC_Ec_, with some completely losing activity, while two variants displayed a twofold decrease in MIC_Bs_. Cytotoxicity remained low, with slight increases in some variants (Figure 7A–C, Appendix A). Thus, Hm-AMP2 variants diverged in properties without overall improvement, reflected in reduced TI values. Hm-AMP2 appears to be a well-balanced AMP, resistant to enhancement by mutations.

Our hybrid approach, combining controlled mutagenesis with intracellular phenotypic screening, serves as a functional platform for the discovery of antimicrobial peptides (AMPs). The results from working with three model AMPs—melittin, cecropin, and Hm-AMP2—demonstrate the feasibility of engineering peptides with enhanced properties. By serving as an empirical alternative to in silico design and a directed evolution strategy beyond standard mutagenesis, it allows for the engineering of peptides with tailored properties. For cytolytic peptides like melittin, the challenge lies not in enhancing their potency, as it is already maximal, but in reducing their toxicity. The mutant MR1P7, with its high therapeutic index, proves the possibility of developing safer versions of potent yet toxic natural AMPs for therapeutic use. Conversely, for peptides such as cecropin, the goal is to broaden their spectrum of antimicrobial activity or increase their efficacy, as evidenced by the emergence of anti-Gram-positive activity in some mutant variants.

The resulting peptides are promising candidates for new therapeutics, both as standalone treatments and in combination with known antibiotics. By disrupting the bacterial membrane, AMPs can facilitate the entry of antibiotics into bacterial cells, restoring their efficacy and overcoming resistance. Such a synergistic approach could significantly extend the useful lifespan of existing antibiotics. Thus, our study provides both a versatile discovery tool and a strategy for designing new peptides to combat antibiotic resistance.

## 4. Materials and Methods

### 4.1. Bacterial Strains and Plasmids

The strain used for recombinant plasmid construction was *E. coli* TOP10 *(F^−^ mcrA Δ(mrr-hsdRMS-mcrBC) φ80lacZΔM15 ΔlacX74 nupG recA1 araD139 Δ(ara-leu)7697 galE15 galK16 rpsL(Str^R) endA1 λ^−^)* (Invitrogen, Carlsbad, CA, USA). The strain used for antimicrobial activity assays was *E. coli* BL21-Gold (DE3) *(F^−^ ompT hsdS(r_B^−^ m_B^−^) dcm^+^ Tet^R gal λ(DE3) endA)* (Novagen, Madison, WI, USA). For testing antimicrobial activity, *E. coli* MG1655 (ATCC, Manassas, VA, USA) and *B. subtilis* 168 HT (was kindly provided by A. Prozorov (Vavilov Institute of General Genetics, Russian Academy of Sciences)) strains were used.

To construct plasmid libraries encoding mutant variants of antimicrobial peptide genes, we used a modified pET22b vector (MilliporeSigma, Burlington, MA, USA). Because insertions or deletions may occur during library generation by hybridization of partially degenerate nucleotides, additional sequences encoding stop codons in alternative reading frames were introduced. The pET22b plasmid was digested with the restriction endonucleases HindIII and XhoI (Thermo Fisher Scientific, Waltham, MA, USA). The oligonucleotides stopF (5′-AGCTTTAAGTAGGCGGCCGCAC-3′) and stopR (5′-TCGAGTGCGGCCGCCTACTTAA-3′) were annealed in ligation buffer. The resulting duplex was ligated into the vector and transformed into *E. coli* TOP10. The presence of the insert was verified using T7 (5′-TAATACGACTCACTATAGGG-3′) and T7t (5′-GCTAGTTATTGCTCAGCGG-3′) primers, followed by sequencing. The resulting plasmid was named pET22bStop. We also used the pETmin plasmid, a modified version of pET22b in which the sequence encoding the *pelB* signal peptide was deleted.

### 4.2. Synthesis of Partially Degenerate Oligonucleotides

As a mutagenesis strategy allowing for controlled mutation rates, we employed the assembly of AMP-coding DNA fragments using partially degenerate oligonucleotides. Oligonucleotide synthesis was carried out by the phosphoramidite method using an eight-channel ASM 800 DNA synthesizer (Biosset, Novosibirsk, Russia) and Glen UnySupport 1000 solid support (Glen Research, Sterling, VA, USA). For degenerate oligonucleotide synthesis, phosphoramidites dT-CE, dC-CE, dA-CE, and dG-CE (Glen Research, Sterling, VA, USA) were dissolved in acetonitrile. Mixed phosphoramidite solutions were then prepared, each containing 88% vol. of the main phosphoramidite and 4% vol. each of the remaining three. Before the synthesis of the final six nucleotides, the process was paused, the mixed phosphoramidite solutions were replaced with single-component (100%) solutions, and synthesis was continued to complete the last six nucleotides. This step ensured the formation of correctly structured sticky ends required for subsequent duplex cloning into the expression vector.

To predict synthesis errors occurring during the construction of genetic fragments with partially degenerate oligonucleotides, we applied a probabilistic model based on the binomial distribution. We investigated two key parameters: oligonucleotide length (coding for HmAMP2 [36 nt], melittin [78 nt], and cecropin [93 nt]) and the accuracy of chemical synthesis, defined as the probability of correct nucleotide incorporation per cycle. Calculations were performed in Python 3.11.4 using the math and pandas libraries. For each parameter combination (length and % of correct phosphoramidites in the mixture), the probability of observing *k* mutations (misincorporations of non-target nucleotides) was calculated using the binomial distribution formula. Following cloning and sequencing of DNA inserts, the actual mutation frequencies were determined and compared with the predicted values.

### 4.3. Library Assembly and Representativeness Assessment

For library assembly, the pET22bStop plasmid, which is a modified version of pET22b, engineered to include a stop codon sequence in all three reading frames, was digested with the restriction endonucleases NcoI and HindIII (Thermo Fisher Scientific, Waltham, MA, USA), and the linear DNA fragment was purified from an agarose gel. Degenerate oligonucleotides (100 pmol) were mixed in 20 µL of 1× T4 DNA ligase buffer (Thermo Fisher Scientific, Waltham, MA, USA). Duplexes were formed by heating the mixture at 95 °C for 2 min followed by gradual cooling to 30 °C over 1 h. In the next step, 80 ng of linearized plasmid DNA was combined with 5 pmol of oligonucleotide duplexes and subjected to ligation. The ligation products were transformed into *E. coli* TOP10 cells and plated on LB agar containing ampicillin (150 mg/L) (neoFroxx, Hesse, Germany).

To assess the representativeness of the library, 96 individual colonies were picked, while the remaining colonies were scraped from the agar surface and transferred into 100 mL of LB medium supplemented with ampicillin (150 mg/L). The culture was incubated for 1.5 h at 37 °C with shaking, after which the biomass was collected by centrifugation, and plasmids were isolated using the QIAGEN Plasmid Midi Kit (Qiagen, Hilden, Germany).

The cell suspension from the 96 picked colonies (prior to scraping the plate) was used as a template for PCR with T7 and T7t primers. The PCR products were sequenced by Sanger sequencing using an 3500xL Genetic Analyzer (Applied Biosystems, Waltham, MA, USA). Library representativeness was evaluated by determining the proportion of clones with different numbers of mutations leading to changes in the amino acid sequence of the encoded peptide.

### 4.4. Antimicrobial Activity Assay, Clone Selection, and Isolation of Plasmids Encoding Mutant AMPs

Antimicrobial activity was tested as previously described [26]. *E. coli* BL21(DE3)Gold cells were transformed with the plasmid library. For controls, cells were transformed with pET-22b-melittin (encoding the wild-type AMP used to generate the library) and with the empty vector pETmin. Transformed cells were plated on LB agar containing ampicillin (150 µg/mL) and glucose (0.5 g/L) (AppliChem, Darmstadt, Germany), and incubated at 30 °C for 21 h. Colonies were transferred into 96-well flat-bottom plates (Wuxi NEST Biotechnology Co., Wuxi, China) containing 200 µL of LB medium supplemented with ampicillin (150 µg/mL) and resuspended thoroughly. For library-transformed cells, a total of four to six 96-well plates were processed. Plates were incubated in a MB100-2A thermoshaker (Hangzhou Allsheng Instruments Co., Hangzhou, China) for 1 h at 37 °C with shaking at 600 rpm.

Subsequently, cultures from the 96-well plates were spotted (5 µL) onto LB agar plates (Amp 150 µg/mL) for preservation, and each well was also subcultured (1:10 dilution) into fresh 96-well flat-bottom plates under two conditions: LB medium without inducer and LB medium containing 0.1 mM IPTG. The final culture volume in each well was 200 µL. Several wells containing sterile medium were included as contamination controls. Plates were incubated in the thermoshaker at 37 °C with shaking at 600 rpm. Optical density at 600 nm (OD_600_) was measured every hour for six time points using an AMR-100 microplate reader (Hangzhou Allsheng Instruments Co., Hangzhou, China). For each measurement, the OD_600_ values of the experimental and control wells were corrected by subtracting the OD_600_ of the sterile LB medium wells.

For each clone, the ratio R_OD_ = OD_–IPTG_/OD_+IPTG_ was calculated, where OD_+IPTG_ represents OD_600_ in wells with IPTG, and OD_–IPTG_ represents OD_600_ in wells without IPTG. Clones with the highest R_OD_ values were selected in this primary screen. Based on amplicon sequencing, unique clones were retained. Plasmids from selected mutant clones were isolated. Antimicrobial activity was then re-tested using the above-described method, as well as by droplet plating serial dilution assay (see Section 4.5).

### 4.5. Antimicrobial Activity Assay Using Agar Plates

As an additional test of antimicrobial activity to compare selected mutant constructs, a droplet plating method of serially diluted cultures was applied, as previously described [26]. Selected clones from the previous step were cultured, plasmid DNA was isolated, and *E. coli* BL21(DE3)Gold cells were transformed with these plasmids. Plates were incubated at 30 °C for 18 h. Single colonies were picked with a pipette tip and transferred into wells of a 96-well flat-bottom plate (Eppendorf, Hamburg, Germany) containing 200 µL LB medium with reduced NaCl, ampicillin, and glucose (0.5 g/L), and thoroughly resuspended. Two colonies were transferred per sample. Plates were incubated in a thermoshaker MB100-2A at 37 °C for 1 h with shaking at 600 rpm.

A series of tenfold dilutions of the cell suspensions was then prepared. For each dilution of each clone, a 10 µL drop was transferred onto the surface of LB agar plates and allowed to absorb completely. In parallel, control plates contained LB agar with 150 µg/mL ampicillin without transcriptional inducer, and LB agar with 0.1 mM IPTG were used. Plates were incubated at 37 °C for 22 h, after which cell growth or its absence was assessed visually. Photographs of the plates were taken using a Scan 1200 imaging system (Interscience, Saint-Nom-la-Bretèche, France).

### 4.6. Analysis of the Impact of Amino Acid Substitutions on Antimicrobial Peptide Activity

The analysis was carried out in three stages using a custom Python 3.11.4 script with the *pandas* (v2.1.3), *matplotlib* (v3.8.1), and *numpy* (v1.26.2) libraries. Datasets for analysis were generated during a series of experimental studies of antimicrobial peptide libraries. Antimicrobial activity of each peptide was initially assessed using kinetic analysis (Section 4.4). Duplicate sequences were removed from the dataset, retaining only unique peptides by averaging activity values across replicates. Unique sequences were then divided into two groups relative to the activity of the reference peptide: the high group included peptides with higher activity than the original AMP, and the low group included peptides with lower activity. All amino acids were classified into four categories: nonpolar (G, A, V, L, I, M, P, F, W), polar (S, T, C, Y, N, Q), positively charged (K, R, H), and negatively charged (D, E).

For each mutated peptide, it was determined whether an amino acid substitution resulted in a change in class (e.g., polar → nonpolar). Positional frequencies of such changes were calculated separately for active and inactive variants. The data were visualized using combined bar charts.

### 4.7. Antimicrobial Activity Testing of Synthetic Peptides

Peptides were synthesized on an automated Liberty Blue synthesizer (CEM Corp., Matthews, NC, USA) using solid-phase Fmoc chemistry with microwave-assisted heating. Two types of resin were selected for peptide synthesis to accommodate different molecular weights. Fmoc-Rink amide aminomethyl-polystyrene resin (ABCR GmbH & Co KG, Karlsruhe, Germany) with a loading capacity of 0.78 mmol/g was used for peptides with a molecular weight of up to 3 kDa. For peptides with a molecular weight exceeding 3 kDa, Fmoc-Rink amide PEG MBHA resin (ABCR GmbH & Co KG, Karlsruhe, Germany) with a loading capacity of 0.48 mmol/g was used. The target products were obtained with a purity of over 95% after purification of the peptides by reversed-phase HPLC on an AKTA Pure 25 M1 system (GE Healthcare Life Sciences, Chicago, IL, USA) equipped with a Zorbax SB-C18 column (Agilent Technologies, Santa Clara, CA, USA) and UV detection [46]. All HPLC-spectra and HRMS-spectra are available at the Appendix A. The antimicrobial activity of synthetic peptides was measured using the standard microdilution method in microtiter plates [49]. Peptides were dissolved in water to a concentration of 1 mM. Cultures of *Bacillus subtilis* 168 HT and *Escherichia coli* MG1655 were grown in a shaker at 30 °C in MHB medium (BD Difco, Thermo Fisher Scientific Inc., Waltham, MA, USA) for 16 h. The resulting cultures were diluted 1:50 in fresh MHB medium and grown in a shaker at 37 °C until reaching mid-logarithmic phase. The bacterial cultures were then further diluted in MHB to a concentration of 10^6^ CFU/mL.

Peptides were serially twofold diluted in MHB in 96-well plates to final concentrations ranging from 100 to 0.2 µM, with a total volume of 50 µL per well. Subsequently, 50 µL of bacterial suspension (10^6^ CFU/mL) was added to each well. Plates were incubated overnight at 37 °C. Each experiment included positive controls (melittin) and negative controls (without peptide). Antimicrobial activity was assessed by measuring the optical density at 600 nm (OD_600_) using a microplate reader AMR-100. The minimal inhibitory concentration (MIC) was defined as the lowest peptide concentration that completely inhibited bacterial growth.

Bacterial growth kinetics were measured in a similar manner. Peptides and bacterial cultures were prepared as described above, but the volume of culture per well was doubled: 100 µL of peptide solution in MHB was mixed with 100 µL of bacterial suspension (10^6^ CFU/mL) in each well. Plates were incubated in a thermoshaker MB100-2A at 37 °C with shaking at 600 rpm for 1 h. OD_600_ was measured hourly for six time points, with an additional seventh measurement after 20 h of incubation.

### 4.8. Cytotoxicity Assay

Peptide cytotoxicity was assessed using the MTT assay on Expi293F cells (human embryonic kidney cells) (Thermo Fisher Scientific, Waltham, MA, USA). Viable cells with active metabolism reduce water-soluble MTT (Abcam, Cambridge, UK) to insoluble formazan, while dead cells lose this ability and produce no signal. The absorbance measured at 600 nm is proportional to the number of living cells. Expi293F cells were seeded in 96-well plates at a density of 1 × 10^5^ cells/mL. After 24 h, peptides were added to the wells at final concentrations ranging from 200 to 0.0975 µM. OptiMEM (Thermo Fisher Scientific, Waltham, MA, USA) without peptides served as a negative control, and melittin at a toxic concentration (100 µM) was used as a positive control. Cells were incubated with peptides in a CO_2_ incubator at 37 °C and 5% CO_2_ for 24 h. After incubation, 10 µL of 0.5 mM MTT was added to each well, and plates were incubated for 4 h at 37 °C, 5% CO_2_. Then, 100 µL of lysis buffer (10% SDS, 0.01 M HCl) was added to each well, and plates were incubated at 37 °C for an additional 16 h. Peptide cytotoxicity was evaluated by measuring optical density at 600 nm and 690 nm using a microplate reader AMR-100. Normalized absorbance was defined as (OD_600_–OD_690_). The percentage of cell viability was calculated using the formula: % Cell viability = (OD_treated_/OD_total_) × 100, where OD_treated_ is the normalized absorbance of treated cells and OD_total_ is the normalized absorbance of non-treated (control) cells.

## Figures and Tables

**Figure 1 antibiotics-14-01120-f001:**
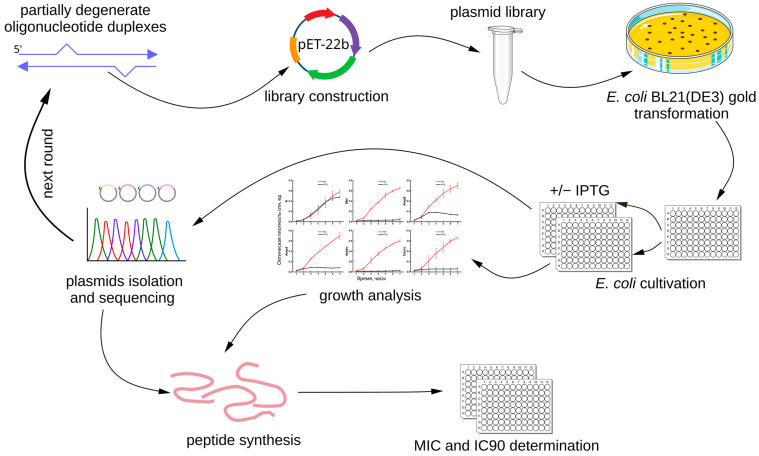
Schematic overview of the experimental approach. A library of expression plasmids encoding random mutants of well-characterized antimicrobial peptides (Hm-AMP2, melittin and cecropin) was generated by random mutagenesis with partially degenerate oligonucleotides. The plasmid library was transformed into *E. coli*, followed by induction of recombinant AMP expression. Clones with the strongest growth inhibition were selected for subsequent rounds of mutagenesis and screening. The most promising variants were synthesized and tested for their properties.

**Figure 2 antibiotics-14-01120-f002:**
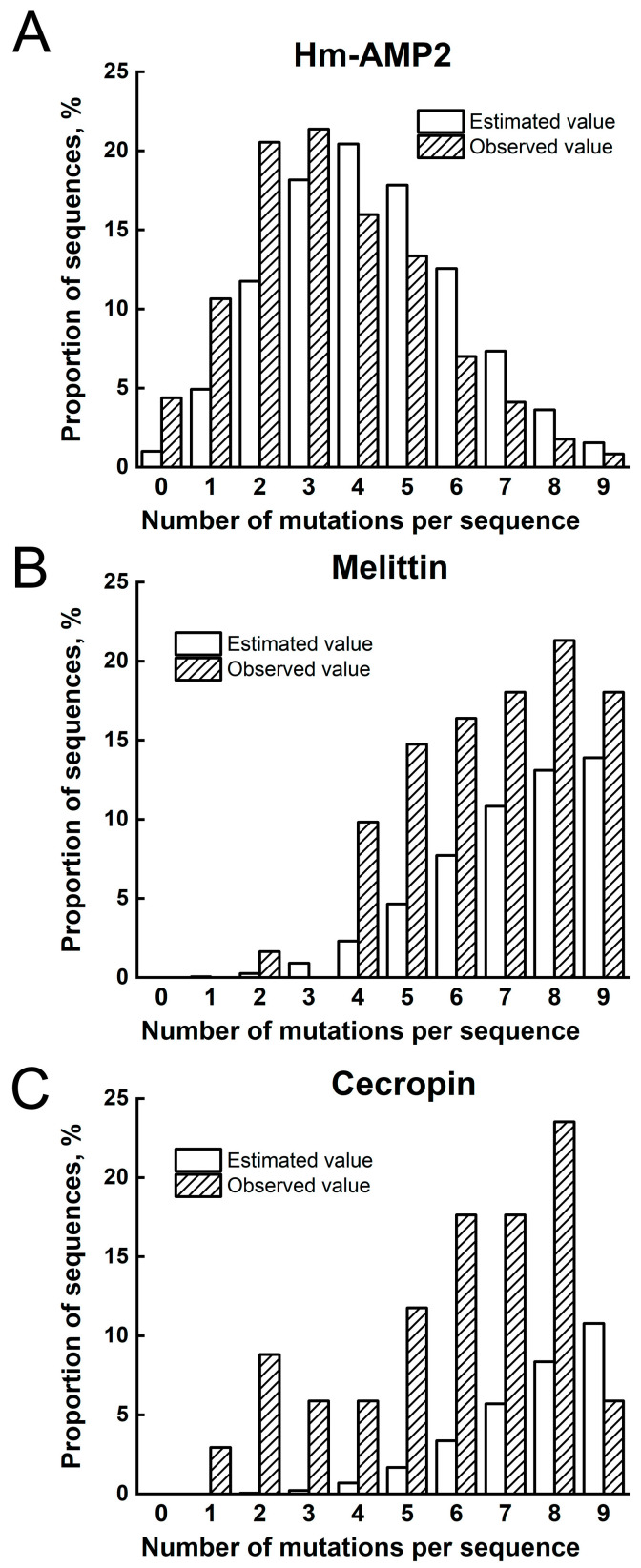
Calculated and observed numbers of mutations for mutant plasmid libraries for (**A**) Hm-AMP2, (**B**) melittin and (**C**) cecropin. *X*-axis: number of nucleotide substitutions; *Y*-axis: fraction of clones with the corresponding number of mutations (%). Estimated value—predicted number of mutations for the “88%” library. This library was synthesized using a phosphoramidite mixture containing 88 vol% of the correct (primary) phosphoramidite and 4 vol% of each of the other three nucleotides at every position.

**Figure 3 antibiotics-14-01120-f003:**
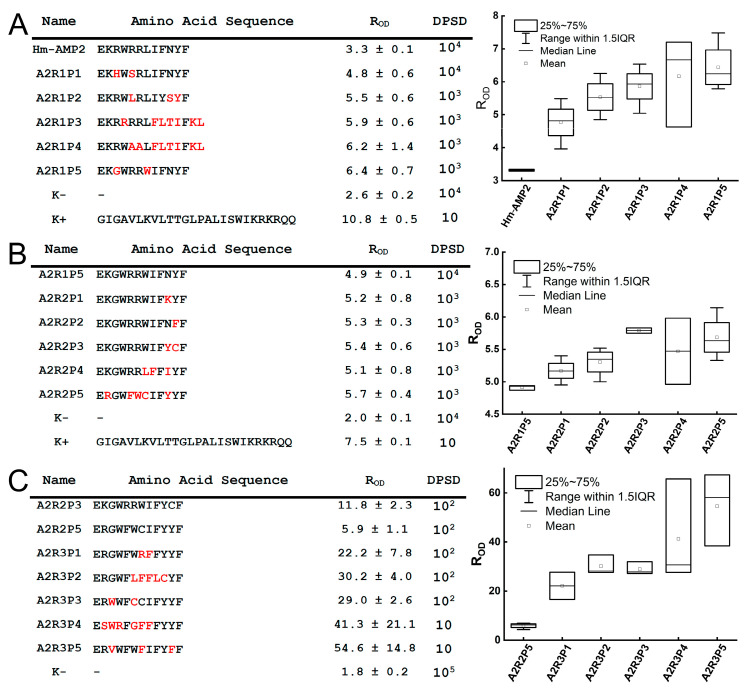
Analysis of the antimicrobial activity of libraries encoding Hm-AMP2 variants. (**A**)—first round of mutagenesis. (**B**)—second round of mutagenesis. (**C**)—third round of mutagenesis. For each round, the five most active peptides were selected. Amino acid substitutions are indicated in red. Activity was assessed using the R_OD_ value and verified by droplet serial dilution. DPSD—Droplet Serial Dilution—the first dilution at which no visible colonies were observed on agar plates. Boxplots show R_OD_ values from experiments evaluating the antibacterial activity of the five most active peptides. The edges of the boxes represent the 25th and 75th percentiles, the line inside the box indicates the median (50th percentile), and the whiskers correspond to the limits of the statistically significant sample. Whisker length is defined as 1.5× the interquartile range below the first quartile and 1.5× above the third quartile. The vertical axis represents R_OD_ values measured 5 h after the start of the kinetic assay.

**Figure 4 antibiotics-14-01120-f004:**
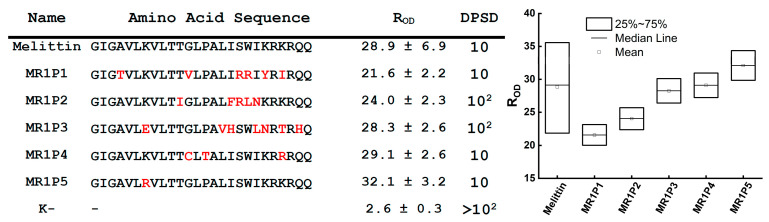
Analysis of the antimicrobial activity of library encoding melittin variants. The five most active peptides were selected. Amino acid substitutions are indicated in red. Activity was evaluated using the R_OD_ value and verified by droplet serial dilution. DPSD—Droplet Serial Dilution—the first dilution at which no visible colonies were observed on agar plates. Boxplots show R_OD_ values from experiments assessing the antibacterial activity of the five most active peptides. The edges of the boxes represent the 25th and 75th percentiles, the line inside the box indicates the median (50th percentile). The vertical axis shows R_OD_ values measured 5 h after the start of the kinetic assay.

**Figure 5 antibiotics-14-01120-f005:**
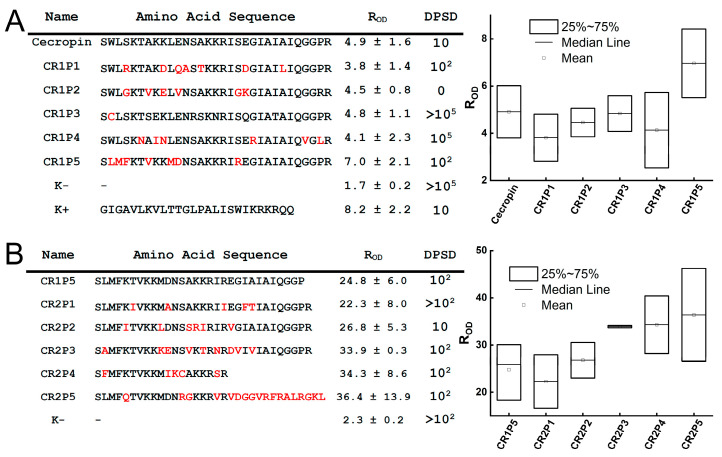
Analysis of the antimicrobial activity of libraries encoding cecropin variants. (**A**)—first round of mutagenesis. (**B**)—second round of mutagenesis. For each round, the five most active peptides were selected. Amino acid substitutions are indicated in red. Activity was evaluated using the R_OD_ value and verified by droplet serial dilution. DPSD—Droplet Serial Dilution—the first dilution at which no visible colonies were observed on agar plates. Boxplots show R_OD_ values from experiments assessing the antibacterial activity of the five most active peptides. The edges of the boxes represent the 25th and 75th percentiles, the line inside the box indicates the median (50th percentile). The vertical axis shows R_OD_ values measured 5 h after the start of the kinetic assay.

**Figure 6 antibiotics-14-01120-f006:**
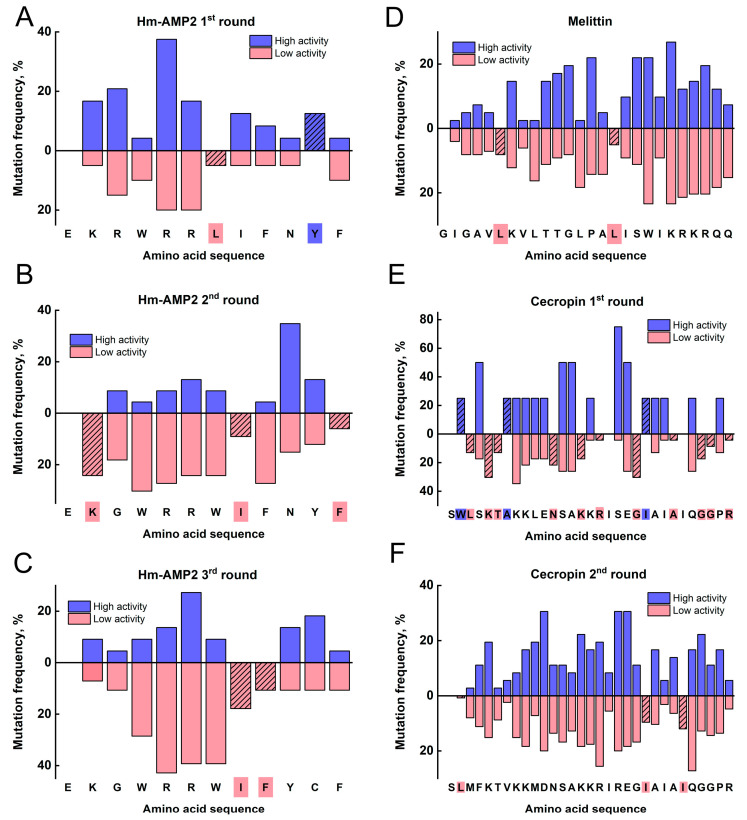
Analysis of position-dependent amino acid substitution frequencies. For each histogram, the frequency of mutations leading to a change in amino acid class is shown for each residue in the sequence of the investigated antimicrobial peptide. Blue indicates substitutions at positions associated with increased activity, while pink indicates substitutions associated with decreased activity. Hatched bars, as well as coloured residues in the peptide sequence, indicate substitutions that occur exclusively in either the more active or the less active peptides. (**A**)—First round of mutagenesis of Hm-AMP2. The original Hm-AMP2 sequence is shown below. (**B**)—Second round of mutagenesis of Hm-AMP2. The sequence of peptide A2R1P5, which showed the highest activity in experiments, is shown below. (**C**)—Third round of mutagenesis of Hm-AMP2. The sequence of peptide A2R2P3, which showed the highest activity in the second round of mutagenesis, is shown below. (**D**)—Results of mutation frequency analysis for melittin mutant libraries. The sequence of melittin is shown below. (**E**)—First round of mutagenesis of cecropin. The original cecropin sequence is shown below. (**F**)—Second round of mutagenesis of cecropin. The sequence of peptide CR1P5, which showed the highest activity in experiments, is shown below.

**Figure 7 antibiotics-14-01120-f007:**
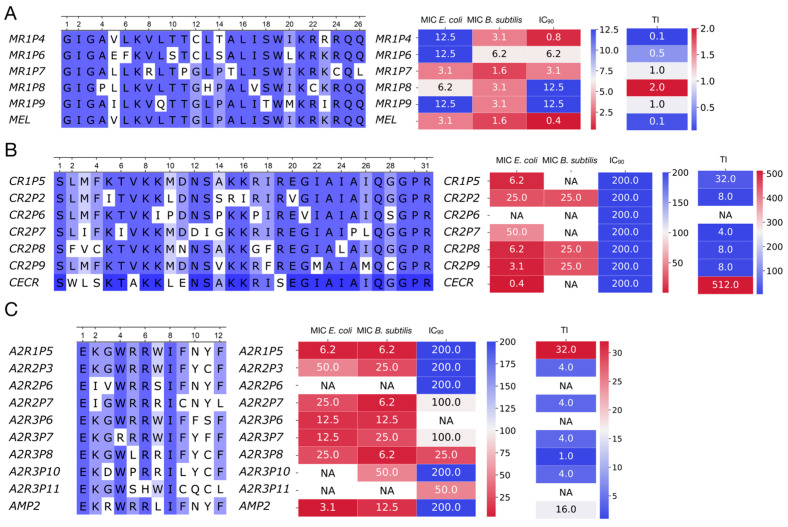
Activity of synthetic peptides melittin (**A**), cecropin (**B**), and Hm-AMP2 (**C**) in relation to introduced mutations. In the alignment on the left, shades of blue indicate amino acid sequence similarities. On the right, antimicrobial activity against *E. coli* and *B. subtilis*, cytotoxicity toward Expi293F cells, and the calculated therapeutic index based on peptide activities are shown. MEL—melittin, CECR—cecropin, AMP2—Hm-AMP2 peptide. AMP’s cytotoxicity was measured as the IC_90_, the concentration that inhibits 90% of cell growth. NA (not applicable) indicates that peptide showed no activity within tested concentration range. Minimal inhibitory concentration (MIC) and IC_90_ are given in µM. The therapeutic index (TI) was defined as the lowest IC_90_ divided by the highest MIC.

## Data Availability

The original contributions presented in this study are included in this article/Appendix A; further inquiries can be directed to the corresponding authors.

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
