# Peer review of "Development of New Antimicrobial Peptides by Directional Selection"

_antibiotics, 2025, doi:10.3390/antibiotics14111120_

Round 1

Reviewer 1 Report

Comments and Suggestions for Authors

In this manuscript, Grafskaia et al. presents a well-structured study aimed at improving the therapeutic potential of antimicrobial peptides (AMPs) through mutagenesis and high-throughput screening. The experimental design is robust, combining mutant library construction, bacterial growth assays, and cytotoxicity profiling to identify optimized variants. Results are clearly described and demonstrate tangible improvements in select peptide candidates with reduced cytotoxicity and broadened antibacterial activity. Overall, this is a valuable and well-executed contribution to peptide-based antimicrobial development; I recommend acceptance pending minor revisions addressing the points below.

  1. The authors state in the text that TI = MIC / Cytotoxicity, but Figure 7 appears to calculate TI as Cytotoxicity / MIC. The conventional and most informative definition is TI = (cytotoxic concentration, e.g. IC90) / MIC. Please correct the manuscript or the figure so the definition is consistent throughout.
  2. It is unclear from Figure 7 whether the reported TI values refer to coli or B. subtilis. Since MICs are provided for both pathogens, TI should be reported separately for each organism or the figure must be clearly labeled to indicate which organism each TI corresponds to.
  3. On line 221, please change figure 3 to figure 6.

Author Response

Comments 1: The authors state in the text that TI = MIC / Cytotoxicity, but Figure 7 appears to calculate TI as Cytotoxicity / MIC. The conventional and most informative definition is TI = (cytotoxic concentration, e.g. IC90) / MIC. Please correct the manuscript or the figure so the definition is consistent throughout.

Response 1: Thank you for pointing out a wrong statement used in the article. We acknowledge the error in the description of the term "therapeutic index," even though the data were calculated correctly and the correct values are provided in the tables/figures. To address this, we have made the following changes:

“... as the ratio of an AMP's cytotoxic concentration to its effective antimicrobial concentration. Results: ...“

“... In our study, it was calculated as the minimum cytotoxic concentration (IC90) divided by the maximum MIC value [25]. For all tested...“

“... The therapeutic index (TI) was defined as the lowest IC90 divided by the highest MIC. ...“

Comments 2: It is unclear from Figure 7 whether the reported TI values refer to coli or B. subtilis. Since MICs are provided for both pathogens, TI should be reported separately for each organism or the figure must be clearly labeled to indicate which organism each TI corresponds to.

Response 2: The therapeutic index (TI), typically defined as the ratio of a drug's toxic concentration to its therapeutic concentration, is used here to evaluate a peptide's potential for broad-spectrum, rather than species-specific, antimicrobial potential. Accordingly, the reported TI represents cytotoxicity to antimicrobial efficacy across several bacterial targets. The description to  Figure 7 has been changed according to the reviewer’s comment.

Comments 3: On line 221, please change figure 3 to figure 6.

Response 3: Corrections have been applied.

Reviewer 2 Report

Comments and Suggestions for Authors

In this manuscript, the authors employed mutagenesis to generate libraries of three antimicrobial peptides - melittin, cecropin, and Hm-AMP2 and evaluated their activity using a growth inhibition assay. Selected mutants were subsequently synthesized to re-test for antimicrobial activity and toxicity. The authors concluded that this strategy yields peptides with an improved therapeutic profile.

The authors provided extensive experimental work, and some of the derivatives showed equal therapeutic efficacy with better toxicity profiles.  The manuscript provides an interesting concept to generate libraries of natural AMP peptides; it lacks the following:

  1. The expression levels and solubility of each peptide mutant significantly affect the activity data. Although the authors acknowledged these limitations, they did not provide any validation experimentation to minimize these limitations.
  2. The analytical characterization of synthetic peptides (HPLC and HRMS) is missing. This data is vital to consider the credibility of the report. Also, detailed synthetic details such as resin type, loading capacity, and purifying yield are needed to replicate the work.
  3. There are no error bars for the MIC and toxicity dataset, which again questions the data measurement and statistical analysis. Moreover, authors also missed the supporting data corresponding to the MIC and toxicity (concentration vs activity/toxicity over time) in addition to providing the numerical values.
  4. The discussion section is poorly written. It primarily focuses on experimental details while also highlighting some limitations of the study. However, do not revisit or validate the claims made in the introduction, such as the alternative to computation design and how it differs from existing mutagenesis strategies.

As stated, the authors should revisit the above-discussed points and make the necessary changes before this work can be considered for publication. Therefore, I recommend rejecting the manuscript in its current state.

Author Response

Comments 1: The expression levels and solubility of each peptide mutant significantly affect the activity data. Although the authors acknowledged these limitations, they did not provide any validation experimentation to minimize these limitations.

Response 1: The experimental validation of peptide presence is inherently unachievable due to the fundamental principles of the method, a limitation that we have acknowledged in the manuscript. Direct peptide detection is impossible: mRNA analysis by real-time PCR is unreliable due to the poor correlation between mRNA andprotein levels in general and because of possible various translation efficacy of the different mRNAs in particular. Immunodetection is unfeasible due to cell death upon expression of active AMPs and the variable stability of the peptides and possible different antigen determinant of mutant peptides. We made the following changes in the revised manuscript in Discussion section:

“... This screening method is highly efficient, enabling the testing of a large number of sequences with moderate resource use, whereas screening a comparable number of synthetic peptides would be far more labor-intensive and costly [26,45,46]. However, a critical interpretation of the results necessitates consideration of the method's key limitations. The primary constraint is the unknown AMP expression level in individual clones, which is influenced by gene structure and peptide stability. This issue arises from the fundamental inability to accurately measure the synthesis and intracellular accumulation of AMPs. Direct detection to quantify expression is not feasible. Detection at the RNA level is unreliable because transcript levels do not reliably correlate with protein levels, and transcripts from mutant libraries exhibit high stochasticity. Similarly, immunodetection is unworkable because high-level expression of active AMPs induces cell death, limiting the peptide available for detection, while the varying intracellular half-lives of different peptides further complicates reliable quantification.

Consequently, the observed changes in growth inhibition (ROD) could be a result of either the true intrinsic antimicrobial activity of the peptides or differences in their expression levels. This means the method does not allow for a direct comparison of the relative activity of the peptides or for the determination of quantitative parameters (MIC, IC90) based on the screening data alone, as the ROD value reflects the combined effect of peptide expression and function. It is also crucial to note the difference between intracellular accumulation during heterologous expression and the extracellular activity of synthetic peptides.

In our study, some of these limitations were mitigated because the compared peptides were of similar length and primary structure. In this regard, the screening data are of a preliminary nature. The final functional profiles of the selected variants (MIC, IC90, TI) were established in subsequent experiments using synthetic peptides in standard extracellular assays. Therefore, while the proposed plasmid-based screening is an effective tool for the primary screening of extensive AMP libraries, its results require mandatory validation by traditional microbiological methods. ...“

Comments 2: The analytical characterization of synthetic peptides (HPLC and HRMS) is missing. This data is vital to consider the credibility of the report. Also, detailed synthetic details such as resin type, loading capacity, and purifying yield are needed to replicate the work.

Response 2:  All HPLC-spectrа and HRMS-spectrа were added to the Supplement material. We used Fmoc-Rink amide aminomethyl-polystyrene resin (ABCR GmbH & Co KG, Karlsruhe, Germany) with a loading capacity of 0.78 mmol/g and Fmoc-Rink amide aminomethyl-polystyrene resin (ABCR GmbH & Co KG, Karlsruhe, Germany),  loading capacity of 0.48 mmol/g. We made the following changes in the revised manuscript:

“...using solid-phase Fmoc chemistry with microwave-assisted heating. Two types of resin were selected for peptide synthesis to accommodate different  molecular weights.  Fmoc-Rink amide aminomethyl-polystyrene resin (ABCR GmbH & Co KG, Karlsruhe, Germany) with a loading capacity of 0.78 mmol/g was used for peptides with a molecular weight of up to 3 kDa.  For peptides with a molecular weight exceeding 3 kDa, Fmoc-Rink amide PEG MBHA resin (ABCR GmbH & Co KG, Karlsruhe, Germany) with a loading capacity of 0.48 mmol/g was used. The target products were obtained with a purity of over 95% after purification of the peptides by reversed-phase HPLC on an AKTA Pure 25 M1 system...“

“...using an MTT assay (Figure 7 A–C, Table S1-S4). The IC90 value, ...“

“... detection [46]. All HPLC-spectrа and HRMS-spectrа are available at the Supplement material (Figures S1-S23). The antimicrobial ...“

Comments 3: There are no error bars for the MIC and toxicity dataset, which again questions the data measurement and statistical analysis. Moreover, authors also missed the supporting data corresponding to the MIC and toxicity (concentration vs activity/toxicity over time) in addition to providing the numerical values.

Response 3: Owing to the discrete nature of the MIC data generated by the microbroth dilution method, statistical error bars are not applicable. The MIC values for all peptides, as well as the processed cytotoxicity data (expressed as mean ± standard deviation), were provided in the Supplementary Material. We made the following changes in the revised manuscript:

“...IC90 (Figure 7 A–C, Table S3), with one variant, ...“

“... variants (Figure 7 A–C, Table S4). Thus, Hm-AMP2, ...“

Comments 4: The discussion section is poorly written. It primarily focuses on experimental details while also highlighting some limitations of the study. However, do not revisit or validate the claims made in the introduction, such as the alternative to computation design and how it differs from existing mutagenesis strategies.

Response 4: Thank you for important remark. We have thoroughly revised the Discussion section and have made every effort to incorporate all of the reviewers' comments. We made the following changes in the revised manuscript in Discussion section:

“...In this study, we proposed a hybrid approach combining the design of new peptides based on known AMPs using mutant libraries with selection in bacterial cells expressing these novel AMPs. Our approach that serves as a functional alternative to purely computational design. Unlike in silico methods that predict a structure before synthesis, our method is empirical and functional. It enables the testing of vast libraries of variants directly inside the bacterial cell, selecting them not for predicted stability, but for their actual antimicrobial activity. A key distinction from standard mutagenesis is that we do not merely create random mutants. Instead, we use an iterative process (rounds of mutagenesis and screening) that mimics natural selection, guiding the evolution of the peptides toward preserving or enhancing their function under physiologically relevant conditions (inside the bacterial cell). ...“

“... This screening method is highly efficient, enabling the testing of a large number of sequences with moderate resource use, whereas screening a comparable number of synthetic peptides would be far more labor-intensive and costly [26,45,46]. However, a critical interpretation of the results necessitates consideration of the method's key limitations. The primary constraint is the unknown AMP expression level in individual clones, which is influenced by gene structure and peptide stability. This issue arises from the fundamental inability to accurately measure the synthesis and intracellular accumulation of AMPs. Direct detection to quantify expression is not feasible. Detection at the RNA level is unreliable because transcript levels do not reliably correlate with protein levels, and transcripts from mutant libraries exhibit high stochasticity. Similarly, immunodetection is unworkable because high-level expression of active AMPs induces cell death, limiting the peptide available for detection, while the varying intracellular half-lives of different peptides further complicates reliable quantification.

Consequently, the observed changes in growth inhibition (ROD) could be a result of either the true intrinsic antimicrobial activity of the peptides or differences in their expression levels. This means the method does not allow for a direct comparison of the relative activity of the peptides or for the determination of quantitative parameters (MIC, IC90) based on the screening data alone, as the ROD value reflects the combined effect of peptide expression and function. It is also crucial to note the difference between intracellular accumulation during heterologous expression and the extracellular activity of synthetic peptides.

In our study, some of these limitations were mitigated because the compared peptides were of similar length and primary structure. In this regard, the screening data are of a preliminary nature. The final functional profiles of the selected variants (MIC, IC90, TI) were established in subsequent experiments using synthetic peptides in standard extracellular assays. Therefore, while the proposed plasmid-based screening is an effective tool for the primary screening of extensive AMP libraries, its results require mandatory validation by traditional microbiological methods. ...“

“...Our hybrid approach, combining controlled mutagenesis with intracellular phenotypic screening, serves as a functional platform for the discovery of antimicrobial peptides (AMPs). The results from working with three model AMPs—melittin, cecropin, and Hm-AMP2—demonstrate the feasibility of engineering peptides with enhanced properties. By serving as an empirical alternative to in silico design and a directed evolution strategy beyond standard mutagenesis, it allows for the engineering of peptides with tailored properties. For cytolytic peptides like melittin, the challenge lies not in enhancing their potency, as it is already maximal, but in reducing their toxicity. The mutant MR1P7, with its high therapeutic index, proves the possibility of developing safer versions of potent yet toxic natural AMPs for therapeutic use. Conversely, for peptides such as cecropin, the goal is to broaden their spectrum of antimicrobial activity or increase their efficacy, as evidenced by the emergence of anti-Gram-positive activity in some mutant variants.

The resulting peptides are promising candidates for new therapeutics, both as standalone treatments and in combination with known antibiotics. By disrupting the bacterial membrane, AMPs can facilitate the entry of antibiotics into bacterial cells, restoring their efficacy and overcoming resistance. Such a synergistic approach could significantly extend the useful lifespan of existing antibiotics. Thus, our study provides both a versatile discovery tool and a strategy for designing new peptides to combat antibiotic resistance. ...“

Reviewer 3 Report

Comments and Suggestions for Authors

In the article “Development of New Antimicrobial Peptides by Directional Selection”, the authors systematically selected antimicrobial peptides by screening mutant libraries, chemically synthesizing these peptides, and demonstrating their antimicrobial properties. The authors have followed a sound scientific approach in designing the experiments, and the results and discussion are well presented.

Minor Comments:

  1. Please include a reference for lines 54–61.
  2. Briefly mention whether there is a difference in the mechanism of action of these AMPs in achieving antibacterial activity, and explain how these AMPs show selectivity against certain bacterial groups.
  3. Include the limitations of the study design and future perspectives, and expand on the potential applications of the current findings in the discussion.
  4. Please provide validation information for the human 541 embryonic kidney cells.

Author Response

Comments 1: Please include a reference for lines 54–61.

Response 1:  We made the following changes in the revised manuscript: “…As a result, many natural peptides evolved to function in environments quite different from those found in the human body [16,17]. Physicochemical conditions in a leech’s mucous cocoon or marine cnidarians differ greatly from human physiological settings. Moreover, the spectrum of microorganisms targeted by innate immunity of arthropods, worms, or cnidarians is distinct from that threaten human health [18,19]. Natural AMPs must also maintain a delicate balance: eliminating pathogens while preserving the stable microbiome [20,21]. Consequently, natural AMPs may represent not the most effective antimicrobial agents, but rather compromise solutions adapted to ecological balance. ...“

Comments 2: Briefly mention whether there is a difference in the mechanism of action of these AMPs in achieving antibacterial activity, and explain how these AMPs show selectivity against certain bacterial groups.

Response 2: Thank you for your valuable suggestion. We have added a brief discussion on these points in the revised manuscript to improve the clarity and depth of our Discussion section: “...  alternative to conventional antibiotics due to their potent efficacy and low potential for cross-resistance [3]. This advantageous profile is underpinned by their fundamental mechanism of action. AMPs initially bind to anionic bacterial surfaces via electrostatic attraction, a step that provides selectivity over mammalian cells [34]. Subsequent bactericidal actions, however, are heterogeneous, involving diverse pathways such as pore formation or detergent-like dissolution of the lipid bilayer, either exclusively or concurrently [35,36]. Two main approaches ...“

Comments 3: Include the limitations of the study design and future perspectives, and expand on the potential applications of the current findings in the discussion.

Response 3: We made the following changes in the revised manuscript in Discussion section: “... This screening method is highly efficient, enabling the testing of a large number of sequences with moderate resource use, whereas screening a comparable number of synthetic peptides would be far more labor-intensive and costly [26,45,46]. However, a critical interpretation of the results necessitates consideration of the method's key limitations. The primary constraint is the unknown AMP expression level in individual clones, which is influenced by gene structure and peptide stability. This issue arises from the fundamental inability to accurately measure the synthesis and intracellular accumulation of AMPs. Direct detection to quantify expression is not feasible. Detection at the RNA level is unreliable because transcript levels do not reliably correlate with protein levels, and transcripts from mutant libraries exhibit high stochasticity. Similarly, immunodetection is unworkable because high-level expression of active AMPs induces cell death, limiting the peptide available for detection, while the varying intracellular half-lives of different peptides further complicates reliable quantification.

Consequently, the observed changes in growth inhibition (ROD) could be a result of either the true intrinsic antimicrobial activity of the peptides or differences in their expression levels. This means the method does not allow for a direct comparison of the relative activity of the peptides or for the determination of quantitative parameters (MIC, IC90) based on the screening data alone, as the ROD value reflects the combined effect of peptide expression and function. It is also crucial to note the difference between intracellular accumulation during heterologous expression and the extracellular activity of synthetic peptides.

In our study, some of these limitations were mitigated because the compared peptides were of similar length and primary structure. In this regard, the screening data are of a preliminary nature. The final functional profiles of the selected variants (MIC, IC90, TI) were established in subsequent experiments using synthetic peptides in standard extracellular assays. Therefore, while the proposed plasmid-based screening is an effective tool for the primary screening of extensive AMP libraries, its results require mandatory validation by traditional microbiological methods. ...“

“... Our hybrid approach, combining controlled mutagenesis with intracellular phenotypic screening, serves as a functional platform for the discovery of antimicrobial peptides (AMPs). The results from working with three model AMPs—melittin, cecropin, and Hm-AMP2—demonstrate the feasibility of engineering peptides with enhanced properties. By serving as an empirical alternative to in silico design and a directed evolution strategy beyond standard mutagenesis, it allows for the engineering of peptides with tailored properties. For cytolytic peptides like melittin, the challenge lies not in enhancing their potency, as it is already maximal, but in reducing their toxicity. The mutant MR1P7, with its high therapeutic index, proves the possibility of developing safer versions of potent yet toxic natural AMPs for therapeutic use. Conversely, for peptides such as cecropin, the goal is to broaden their spectrum of antimicrobial activity or increase their efficacy, as evidenced by the emergence of anti-Gram-positive activity in some mutant variants.

The resulting peptides are promising candidates for new therapeutics, both as standalone treatments and in combination with known antibiotics. By disrupting the bacterial membrane, AMPs can facilitate the entry of antibiotics into bacterial cells, restoring their efficacy and overcoming resistance. Such a synergistic approach could significantly extend the useful lifespan of existing antibiotics. Thus, our study provides both a versatile discovery tool and a strategy for designing new peptides to combat antibiotic resistance. ...“

Comments 4: Please provide validation information for the human 541 embryonic kidney cells.

Response 4: We used commercially available cell line Expi293F cells (human embryonic kidney cells) (Thermo Fisher Scientific, USA). The specification can be found at the link: https://www.thermofisher.com/order/catalog/product/A14527

Reviewer 4 Report

Comments and Suggestions for Authors

In this manuscript, the authors generated a screening method for making mutant antimicrobial peptides in E. coli. My comments are as follows:

  1. Page 4, line 96: pET22bStop. In the beginning, I thought it’s a typo. Until page 13, line 406, this plasmid is defined here. For the benefit of readers, the authors might consider to add more details at its first appearance.

  1. Page 11, line 307: physical-chemical or physico-chemical?

  1. Page 11, lines 310-312: One sentence as one paragraph is weired.

  1. Page 11, Material and method section: Please add city name in between company and country. For example, Invitrogen, USA (line 390).

  1. Page 15, line 531: Please provide company, city, and country names for AMR-100.

  1. Page 16, lines 569-574: Please correct/rewrite this paragraph.

  1. Page 17, Abbreviation section: More abbreviations should be included such as MEL (melittin), CECR (cecropin), and so on. Please rearrange it by alphabetical order.

  1. Reference section: Bacteria names should be italic, e.g., Escherichia coli (line 628).

Author Response

Comments 1: In this manuscript, the authors generated a screening method for making mutant antimicrobial peptides in E. coli. My comments are as follows: Page 4, line 96: pET22bStop. In the beginning, I thought it’s a typo. Until page 13, line 406, this plasmid is defined here. For the benefit of readers, the authors might consider to add more details at its first appearance.

Response 1:  The pET22bStop plasmid is a modified version of pET22b. In this variant, the native DNA region encoding stop codon was extended to form a stop codon that functions in all three reading frames. Thus, in the event of a frameshift, it would not result in a mutant peptide longer than the original. We made the following changes in the revised manuscript:

“...pET22bStop plasmid. In this modified version of pET22b, the native DNA region encoding stop codon was extended to form a stop codon that functions in all three reading frames. Thus, in the event of a frameshift, it would not result in a mutant peptide longer than the original. Partially degenerate ...“

“... pET22bStop plasmid, which is a modified version of pET22b, engineered to include a stop codon sequence in all three reading frames, was digested with the restriction ...“

Comments 2: Page 11, line 307: physical-chemical or physico-chemical?

Response 2: We made the relevant corrections.

Comments 3: Page 11, lines 310-312: One sentence as one paragraph is weired.

Response 3: Corrections have been applied.

Comments 4: Page 11, Material and method section: Please add city name in between company and country. For example, Invitrogen, USA (line 390).

Response 4: We made the relevant corrections.

Comments 5: Page 15, line 531: Please provide company, city, and country names for AMR-100.

Response 5: We made the following changes in the revised manuscript:

“... AMR-100 microplate reader (Hangzhou Allsheng Instruments Co., Hangzhou, China). ...“

Comments 6: Page 16, lines 569-574: Please correct/rewrite this paragraph.

Response 6: We made the following changes in the revised manuscript:

“... The original contributions presented in the study are included in the article/Supplementary Material, further inquiries can be directed to the corresponding authors. ...“

Comments 7: Page 17, Abbreviation section: More abbreviations should be included such as MEL (melittin), CECR (cecropin), and so on. Please rearrange it by alphabetical order.

Response 7: We rearranged and widened the list of abbreviations which are used in this manuscript:

“... The following abbreviations are used in this manuscript:

Amp

Ampicillin

AMP

Antimicrobial peptide

AMP2

Hm-AMP2 peptide

CECR

Cecropin

DNA

Deoxyribonucleic acid

DPSD

Droplet serial dilution assay

IC90

Viability index

IPTG

Isopropyl β-d-1-thiogalactopyranoside

LB

Luria-Bertani Broth

MEL

Melittin

MHB

Mueller-Hinton Broth

MIC

Minimal inhibitory concentration

MICBs

Minimal inhibitory concentration against B. subtilis

MICEc

Minimal inhibitory concentration against E. coli

MTT

3-(4,5-dimethylthiazol-2-yl)-2,5-diphenyltetrazolium bromide

PCR

Polymerase chain reaction

TI

Therapeutic index

OD600

Optical density at 600 nm

...“

Comments 8: Reference section: Bacteria names should be italic, e.g., Escherichia coli (line 628).

Response 8: Corrections have been applied.

Round 2

Reviewer 2 Report

Comments and Suggestions for Authors

The revised manuscript is suitable for publication.